# A Customisable Data Acquisition System for Open-Source Hyperspectral Imaging

**DOI:** 10.3390/s23208622

**Published:** 2023-10-21

**Authors:** Yiwei Mao, Christopher H. Betters, Samuel Garske, Jeremy Randle, K. C. Wong, Iver H. Cairns, Bradley J. Evans

**Affiliations:** 1School of Physics, The University of Sydney, Sydney, NSW 2006, Australia; yiwei.mao@sydney.edu.au (Y.M.); christopher.betters@sydney.edu.au (C.H.B.); 2ARC Training Centre for CubeSats UAVs and Their Applications, The University of Sydney, Sydney, NSW 2006, Australia; sam.garske@sydney.edu.au (S.G.); kc.wong@sydney.edu.au (K.C.W.); bradley.evans@une.edu.au (B.J.E.); 3Australian Centre for Field Robotics, The University of Sydney, Sydney, NSW 2006, Australia; jag@acfr.usyd.edu.au; 4School of Environment and Rural Science, University of New England, Armidale, NSW 2351, Australia

**Keywords:** data acquisition, hyperspectral imaging, remote sensing, open-source

## Abstract

Hyperspectral imagers, or imaging spectrometers, are used in many remote sensing environmental studies in fields such as agriculture, forestry, geology, and hydrology. In recent years, compact hyperspectral imagers were developed using commercial-off-the-shelf components, but there are not yet any off-the-shelf data acquisition systems on the market to deploy them. The lack of a self-contained data acquisition system with navigation sensors is a challenge that needs to be overcome to successfully deploy these sensors on remote platforms such as drones and aircraft. Our work is the first successful attempt to deploy an entirely open-source system that is able to collect hyperspectral and navigation data concurrently for direct georeferencing. In this paper, we describe a low-cost, lightweight, and deployable data acquisition device for the open-source hyperspectral imager (OpenHSI). We utilised commercial-off-the-shelf hardware and open-source software to create a compact data acquisition device that can be easily transported and deployed. The device includes a microcontroller and a custom-designed PCB board to interface with ancillary sensors and a Raspberry Pi 4B/NVIDIA Jetson. We demonstrated our data acquisition system on a Matrice M600 drone at a beach in Sydney, Australia, collecting timestamped hyperspectral, navigation, and orientation data in parallel. Using the navigation and orientation data, the hyperspectral data were georeferenced. While the entire system including the pushbroom hyperspectral imager and housing weighed 735 g, it was designed to be easy to assemble and modify. This low-cost, customisable, deployable data acquisition system provides a cost-effective solution for the remote sensing of hyperspectral data for everyone.

## 1. Introduction

Hyperspectral imagers, also known as imaging spectrometers, capture imagery in many narrow spectrally contiguous bands. For remote sensing, this presents two challenges: recording high-sample-rate image data and having the ancillary geospatial information to map them. This ability is well-suited for environmental studies and facilitates measurements of the physical properties of objects without physical contact. Imaging spectroscopy is used in mineral mapping [1,2,3,4,5], geology and soils [6,7,8,9], plant ecology and invasive species monitoring [10,11,12,13,14], hydrology [15,16,17,18,19,20], and the atmospheric modelling of pollutants [21,22,23,24,25]. Many remote sensing applications rely on these instruments deployed on satellites, such as Hyperion onboard the Earth Observing-1 (EO-1) satellite [26,27,28], and mounted on aircraft such as the Airborne Visible/Infrared Imaging Spectrometer (AVIRIS) [29,30,31] and HyMAP [32,33,34].

Recent developments in hyperspectral imager design have made portable low-cost applications possible for studies on leaf composition [35,36], quality control in production [37], aurora observations [38], and urban monitoring [39]. For these compact hyperspectral imagers, sometimes known as do-it-yourself hyperspectral imagers [40,41,42], pushbroom designs are the most popular, with a few using a snapshot design [43]. Diffraction gratings are commonly used [44], although some use prisms as the dispersing mechanism [45]. Some of these can send visualisations to a mobile device using WiFi [35,46], but this is short range by design.

These devices need to be tethered in order to be powered and operated [47], limiting the available applications and preventing true remote sensing onboard autonomous platforms. Substituting for a drone, Ref. [48] deployed one of these hyperspectral imagers from a mast on a taut rope. They did not collect navigation data, so any motion artefacts were not corrected for. With the exception of [49], all of these custom hyperspectral imaging systems have not been deployed on an autonomous platform. In a review by Stuart et al. on the state of compact field deployable hyperspectral imaging systems [50], the complexity of georeferencing was highlighted, as this requires navigation and orientation data to be collected and time-synchronised to the hyperspectral imagery. The availability of a customisable data acquisition system that is capable of collecting hyperspectral, navigation, and orientation data is holding back the deployment of non-commercial hyperspectral imagers for remote sensing.

Despite being out of reach for most remote sensing practitioners and requiring a larger budget, it is worth mentioning that hyperspectral imagers built with COTS components were proposed for CubeSats [51,52,53,54].

Having a standalone data acquisition system to collect navigation and orientation data has a few advantages over extraction from DJI or ArduPilot data logs. The hyperspectral data and ancillary navigation data need to be timestamped and synchronised for accurate georeferencing [55], since time offsets lead to spatial offsets. Therefore, we used the global navigation satellite system (GNSS) time and pulse per second (PPS) signal to synchronise the onboard computer system time. Secondly, in order to accurately model the camera dynamics, the IMU sensor sample rate needs to be faster than the camera frame rate, and this is difficult to control or requires advanced modifications onboard DJI or Ardupilot drones. A customisable data acquisition system allows flexibility in selecting frame rates and exposure times. Furthermore, we were able to measure the orientation of the hyperspectral imager directly by mounting an IMU on the camera body, rather than estimating the orientation by proxy from the UAV or gimbal orientation. These advantages allow our data acquisition system to be deployed beyond UAVs and be used on aircraft and satellites.

While some commercial hyperspectral imagers come with built-in data acquisition (DAQ) systems, these can be prohibitively expensive and may not be suitable for real-time analysis. The cost of one of these data acquisition systems from manufactures such as Headwall, Resonon, or HySpex is not published but is commensurate with the cost of their hyperspectral imagers. The total cost of our data acquisition system is shown in the bill of materials table (Appendix A) and is estimated to be around an order of magnitude more affordable.

Our data acquisition system uses the open-source hyperspectral imager (OpenHSI), which is a compact and lightweight alternative that can be interfaced with development boards [56]. An open-source initiative lowers the barrier to entry for remote sensing practitioners to extend lab-based applications to the field. It also enables contributors to the project to become more involved in the fundamentals of hyperspectral imaging, including calibration, imagery alignment, and spectral data processing, because there are no proprietary barriers or a fixed system design and settings. The data acquisition system and OpenHSI camera combination is a self-contained solution that can be deployed on drones, aircraft, gantries, and tripods. Compared to pre-existing systems, this relies only on commercial-off-the-shelf (COTS) components, a PCB that can be ordered, and 3D printing. Furthermore, a custom data acquisition solution offers opportunities to interact with sensor data during collection.

The work described in this paper is the first attempt at producing a true open-source solution for obtaining remotely sensed hyperspectral data. In this paper, we describe the design and methods needed to replicate the system. We include instructions for the build and operation and conclude with some initial results of our in-flight tests. We tested our DAQ and hyperspectral imager on a drone flight in Sydney, Australia, showing that hyperspectral, navigation, and orientation data can be collected and timestamped concurrently.

## 2. The Data Acquisition System

The OpenHSI camera and data acquisition system is a self-contained package that is designed to be deployed on a drone or aircraft [56] and is capable of collecting the camera position and orientation during flight for later georeferencing. Since the hyperspectral camera is a pushbroom sensor, the drone’s forward motion provides the second spatial dimension. The specifications for the camera are shown in Table 1. The calibration and field validation of the OpenHSI camera design are described in [56]. Iterating on the original design, we used a FLIR detector for the data collected in this work. The open-source Python library used to operate the OpenHSI cameras works with a variety of detector manufacturers, including XIMEA, FLIR, and LUCID.

The data acquisition system is arranged in a stack consisting of a compute board (Raspberry Pi 4B or NVIDIA Jetson), an uninterruptible power supply (UPS) and onboard batteries, and a Printed Circuit Board (PCB) containing the necessary ancillary sensors for direct georeferencing. Connected to the compute board by USB cables are a fast Solid State Device (SSD) for storage and the OpenHSI camera. (For the NVIDIA Jetson version, an M.2 slot SSD can be used instead). Through the General Purpose Input/Output (GPIO) pins, the ancillary sensor data are recorded in parallel with the hyperspectral data. A latching button is used to control the data collection, and LEDs indicate the system’s status. Sensor data are processed and timestamped by a Teensy 4.0 microcontroller before being sent over UART to be saved by the Raspberry Pi. If desired, this datastream can be processed in real-time.

Raspberry Pis have been used in other remote sensing data acquisition tasks. For example, Belcore et al. used an RGB and Infrared Raspberry Pi camera on a drone to capture multispectral imagery [57]. For other lab-based hyperspectral tasks, Nasila et al. used a Raspberry Pi Zero W to interface a custom hyperspectral imager to a wireless network [46].

We found the a Raspberry Pi 4B was sufficient for slow tasks due to the limitations in processing power and USB data bus speeds—a tradeoff for the low cost. This was confirmed by [35], who found that a Raspberry Pi was insufficient for their needs. Hasler et al. successfully deployed a comparable hyperspectral imager controlled by a Jetson TX2 on a drone [49] and collected timestamped sensor data using a similar ancillary sensor board [55] for direct georeferencing. While the authors provided some tips and code listings, the full source and documentation were not available.

The main features of our DAQ are

Entirely self-contained, including power;Navigation and orientation data for georeferencing collected in parallel;Uses commercial off-the-shelf components and 3D printing;Completely open-source with all design files and code available;Can be deployed remotely on drones and aircraft; and An easy-to-use interface (power on, press a button, and go).

Compared to closed-source commercial hyperspectral imagers, in terms of cost, the complete OpenHSI camera and data acquisition system is at least an order of magnitude more affordable. We chose SparkFun and Adafruit for most of our components, because they are reputable suppliers of sensors who also open-source their PCB designs and supporting software. Furthermore, we used an Arduino-compatible microcontroller and built upon the existing open-source code available. While the hardware design described in this paper is specific to the Raspberry Pi, a Jetson board can be used as a drop-in replacement at a higher cost. The GPIO requirements and software are the same for either option.

### 2.1. Electrical Components

In order to make it easy to interface with existing Raspberry Pi hats (accessory boards), the stacks can be connected using the 40 pin GPIO header. All ancillary sensors are controlled by a Teensy 4.0 microcontroller. The reason for including this microcontroller, rather than using the Raspberry Pi directly, is threefold: first, to obtain accurate timestamped sensor data without the delay caused by the operating system scheduler; second, to reduce the overhead for the Raspberry Pi in reading sensor data; and third, we needed two hardware serial ports, whereas the Raspberry Pi only has one. When we initially tried to read the sensor data directly with the Raspberry Pi, the camera frame rate dropped by more than half. The second hardware serial port allowed us to send selective diagnostic messages wirelessly using the XBee wireless radio module.

The board contains a GNSS module, a pressure/humidity/temperature sensor, a real time clock, connectors for the IMU, which is mounted on the camera, a wireless radio, and connectors for a latching button that controls data collection. There are also LED status indicators and a reset button to manually restart the microcontroller. The microcontroller runs a cooperative scheduler that schedules each sensor-read task at its desired frequency. Capacitors are used to stabilise the power and ground planes.

Each component is mounted on pin headers, and each PCB layer stacked, so that users can swap and remove modules as needed for a variety of deployment scenarios and applications. For example, if the XBee is physically removed for a drone flight, the rest of the software still runs but no longer initialises and schedules the Xbee task. This design also includes an independent uninterruptible power supply with 18,650 batteries, removing the reliance on buck converters and the host vehicles’ (drones) power, which will reduce the flight time. Using the onboard USB-C port, the batteries can be charged, with an LED bar indicating the charge level.

A complete bill of materials for our DAQ system can be found in Appendix A. While these components can be sourced from the listed supplier, it is often more convenient to source them locally. Here in Australia, we purchased all of the components listed in Table A1 other than the UPS, SSD, and custom PCB from [58].

Resistors and capacitors are common components to buy in bulk, which reduces the cost for a project that only uses a few. With the PCBs, other fabrication houses will be suitable, but we used the suppliers JLCPCB and OSHPark. We chose black soldermask, but note that certain colours (green) will have faster turnaround times.

Since the data acquisition system and camera are designed to be deployed in a Ronin gimbal (or fixed mounted), the GNSS antenna needs to be brought out and fixed to the top of the drone to provide a direct line of sight. The displacement from the GNSS chip should be measured. The same microcontroller code can be used for the Sparkfun GPS RTK2 if RTK functionality is desired.

We chose the Teensy 4.0 microcontroller because of its many hardware serial ports and compatibility with the Adafruit and Sparkfun sensor libraries. We decided to mount every sensor in headers for ease of replacement and development. This proved to be helpful, as we accidentally blew some capacitors during testing.

The XBEE 3, XBEE USB, and XBEE Breakout listed in Table A1 are optional components. These components form a wireless serial pair, so the the DAQ status can be viewed in real-time during operation. Unfortunately, these wireless modules can only be used in the lab, since the signal is overpowered by the Matrice M600 drone when powered on. This can be rectified using other radios that operate at a different frequency.

The Raspberry Pi 4B, battery hat, and ancillary sensor board are stacked and mounted within the 3D printed housing. Figure 1 shows the assembled DAQ.

### 2.2. Design Files and Assembly

Each component for our DAQ is a commercial-off-the-shelf product and can be either ordered or 3D-printed. For the PCB, we provide the design files needed to order them from PCB fabrication houses. All design files and code can be found at [59]. The entire data acquisition system is mounted within a 3D-printed enclosure with holes for attaching mounting brackets and for airflow. To obtain the appropriate strength, our printer used a wall size of 2 mm and 100% infill.

Assembly is straightforward and involves stacking the Raspberry Pi 4B, battery hat, and ancillary sensor board with spacers to fix the structure in place. The battery hat includes the spacers needed, and we used some Adafruit nylon spacers for the ancillary sensor board. After the stack is complete, the entire system can be fixed to the 3D-printed enclosure with some standoffs. In Figure 2a, a visualisation of the assembly is shown. The overall dimensions of the housing are shown in Figure 2b. The enclosure also has holes to mount it directly to a dovetail bracket, allowing it to be deployed on a Ronin gimbal for our drone flights. Using some foam padding, we placed the SSD inside and connected it via USB. On the side of the 3D-printed enclosure are mounting points for attaching a bracket and the OpenHSI camera. The IMU is attached to a bracket and mounted on the side of the OpenHSI camera. Since the drone body interferes with the GNSS signal, the antenna needs to be carefully routed and fixed to the top of the drone.

The entire setup weighed 735 g as mounted on a drone. There were a few things that could reduce the weight, such as using an SSD without the enclosure for the compute boards with an included M.2 slot. The battery hat could be removed if power could be sourced from the drone or aircraft. The ancillary sensors could also be soldered directly on, as opposed to using headers. This would reduce the weight down to 515 g. This payload is light enough for the DJI Phantom 4 carry capacity.

Once the hardware procedure is built, some software tweaks are needed. Some of the key steps are to enable GPIO, UART, and I2C and then add the current user to the dialout group. We also found it helpful to increase the USB memory limit and use PI0UART. The battery hat (X728) also has its own software [60] that needs to be installed. Once this was complete, we ran a script on power up using systemd. Similar steps are needed to install the software for the Jetson series of development boards. For more details on the software installation, visit our installation guide on our documentation website [61].

## 3. Operating Instructions

Before the DAQ can be deployed remotely, the batteries need to be charged. In our testing, the DAQ was capable of at least 2 h of data collection, which is significantly longer than the drone flight.

Although not necessary, we found that a gimbal greatly improved the data quality on our drone flights. The starting orientation of the camera should be recorded when the DAQ is powered on as this is needed for georectification. The camera can be assumed to be pointing directly down and rotated to a particular heading from geographic north.

A latching button (with an internal LED) is used to control the operation of the DAQ and camera, and a separate button powers the DAQ on and off. This interface was intentionally designed to be as simple as possible. LED status indicators provide information about the operation of the Raspberry Pi and ancillary sensor board stack. Before takeoff, a GNSS fix and the IMU calibration can be checked by observing the heartbeat LED blinking faster than 1 Hz. A GNSS fix can take a few minutes from a cold start. This is summarised in Table 2. For more detailed sensor information, a laptop can parse the XBEE wireless messages in real time and show plots of sensor data on a dashboard. Unfortunately, the M600 drone drowned out the signal when powered up so this could only be used for preflight testing.

After the DAQ is powered on, pressing the LED button begins the data collection, and depressing the button stops the data collection. When the LED indicator stops blinking, the navigation and hyperspectral data have finished saving to an onboard SSD. At this point, the SSD cable can then be unplugged from the Raspberry Pi or Jetson, and summary plots can be viewed to confirm the data were collected correctly.

The operating procedures are summarised in Figure 3.

For spectral validation and empirical line calibration, known spectral targets are beneficial. These can be calibration tarps or colour panels. A combination of a dark and a bright target with known spectral measurements is best. For verification of the georeferencing, a Trimble (or another comparable device) can be used to mark the tarp corners and other targets of interest in view.

There are a few considerations needed for obtaining high quality hyperspectral data. In order to obtain square pixels, the drone or aircraft needs to be flown at a specific speed. If this is not achieved, each pixel will be elongated into a rectangle. The parameters that control the flight speed are the camera frame rate and altitude.

### Obtaining Square Pixels

The frame rate or frames-per-second (FPS) depends heavily on the capability of the data acquisition device. The USB throughput of the Raspberry Pi is a limiting factor for the DAQ, and because it is multiplexed, one cannot operate the camera and save the data to the SSD concurrently. However, this may not necessarily be an issue for some applications. We found the Jetson to be a superior compute platform since it does not have this issue, and hyperspectral data can be captured and saved with minimal latency. In this application, the fastest frame rate we could achieve on a Raspberry Pi was 25 FPS, so we used an integration time of 30 ms to maximise the signal-to-noise ratio without slowing the frame rate any further.

At an altitude *l* of 120 m and a cross track angular resolution δ of 0.24 mrad, the ground sample distance (GSD) is approximately
(1)GSD=ltan(δ)≈lδ≈2.9cm.

Therefore, in order to image square pixels, the drone or aircraft needs to fly at a speed of
(2)v=GSD×FPS≈72cm/s.

See Figure 4 for a visual. Whether the drone or aircraft is able to fly at this speed depends on the platform and control parameters. For example, when using the Pix4D application to control the drone’s flight path, the smallest allowable speed was 1 m/s, so we flew at 1.4 m/s (an integer multiple of *v*) and later spatially binned the data to 5.8 cm pixels. For higher spatial resolutions, the altitude *l* can be decreased accordingly.

## 4. Validation and Characterisation

We validated the DAQ on an Matrice M600 drone and collected hyperspectral data and navigation data simultaneously. Figure 5 shows the beach area where we setup tarps and cones to image with our completed DAQ mounted on a Ronin gimbal and deployed on a Matrice M600 drone.

A summary of the data is shown in Figure 6. From the quaternions, the camera orientation was calculated in a local north-east-down (NED) frame. After every data collection session, the timestamped sensor measurements were saved, along with summary plots.

Figure 7 demonstrates the Raspberry Pi 4B recording the UTC timestamps for each line recorded by the hyperspectral imager during the drone’s forward motion. These timestamps were synced to the microcontroller’s GNSS time using the pulse-per-second signal.

The navigation and orientation data collected after time synchronisation were used to directly georeference the hyperspectral data collected. The resulting datacube can be visualised in Figure 8.

### Usage Tips and Future Work

Without an RTK, the GNSS points were accurate to 1 m, and the maximum drift was observed while the data acquisition system was stationary. While this may not be an issue, depending on the spatial resolution required, the OpenHSI camera had a ground sampling distance of 5 cm, so this was significant. For the highest georeferencing accuracy, an RTK module is recommended. Future trials should collect ground control points so that the accuracy of the georeferenced positions of the cones and tarpaulins can be quantified.

At first startup, the data acquisition system requires time for a GNSS fix and IMU calibration. This is needed before takeoff and is indicated by an LED. Once the IMU has been calibrated, the starting orientation of the camera should be noted, as it is used as the initial condition for georectification. We found gimbal stabilisation helpful for increasing the quality of the georeferenced hyperspectral data, but it is not strictly necessary.

While we validated the data acquisition system on a successful flight, it is worth testing the performance under different wind conditions and without gimbal stabilisation. Doing so will inform users about the structural dynamics and suitability of the ancillary sensor data. This may mean setting constraints on deployment conditions. For example, if there are significant vibrations above 50 Hz or the hyperspectral imager needs to operate at s higher than 50 Hz frame rate, a different IMU with a higher than 100 Hz sample rate is needed.

Despite being affordable, the Raspberry Pi 4B has a multiplexed USB bus, so camera data pause while writing to a portable SSD using the same USB bus. Knowing that there is minimal latency during hyperspectral data collection would give more control over how to best image the area of interest and plan swath overlaps. We found the NVIDIA Jetson boards to be superior computing device, and our open-source software was designed to be compatible with these. Please see [61] for an installation guide. Other options include the Banana Pi, which has options for an M.2 SSD slot and more RAM. These options forgo the portable SSD in favour of an installed SSD module, thereby reducing the weight and overall size of the data acquisition system.

## 5. Conclusions

Previous uses of compact do-it-yourself hyperspectral imagers required the camera to be tethered, limiting the scope to lab-based or handheld applications. For remote sensing applications, a self-contained data acquisition system that is capable of recording hyperspectral and navigation data simultaneously is necessary. This work fills the gap and introduces a compact and customisable open-source data acquisition system for collecting time synchronised hyperspectral data alongside navigation and orientation data for direct georeferencing.

Our hardware design and software solution use only COTS components, a PCB that can be ordered, and a 3D-printed enclosure. A core design decision was to use easily replaceable components and allow ease of operation at the cost of some weight and height. The entire assembly weighed 735 g. We used a Raspberry Pi 4B to control the OpenHSI camera, and we found that the USB bus contention limited the maximum frame rate. A slightly more costly solution was to use a Jetson as a drop-in replacement.

By releasing our data acquisition system designs and code to the open-source community, we have lowered the barrier of entry for remote sensing practitioners to produce their own georeferenced hyperspectral imagery. This also enables contributors to be more involved in the fundamentals of hyperspectral imaging, including calibration, imagery alignment, spectral data processing, and georectification, because there are no proprietary barriers. With this work, we offer a more affordable option for the global community to build an entirely open-source data acquisition system and hyperspectral imager for remote sensing using commercial-off-the-shelf components. Overall, we demonstrated our open-source hardware design and software on a drone flight over Sydney, Australia, proving that hyperspectral remote sensing can be possible at low cost for everyone.

## Figures and Tables

**Figure 1 sensors-23-08622-f001:**
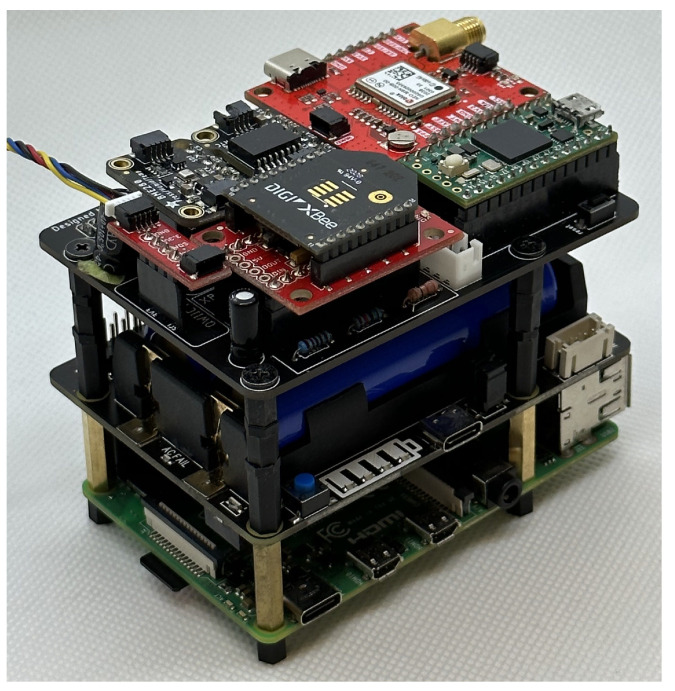
The assembled DAQ including the ancillary sensors. The IMU is connected via a cable so it can be mounted on the camera. The mass is 583 g including the SSD and camera, but not including the enclosure.

**Figure 2 sensors-23-08622-f002:**
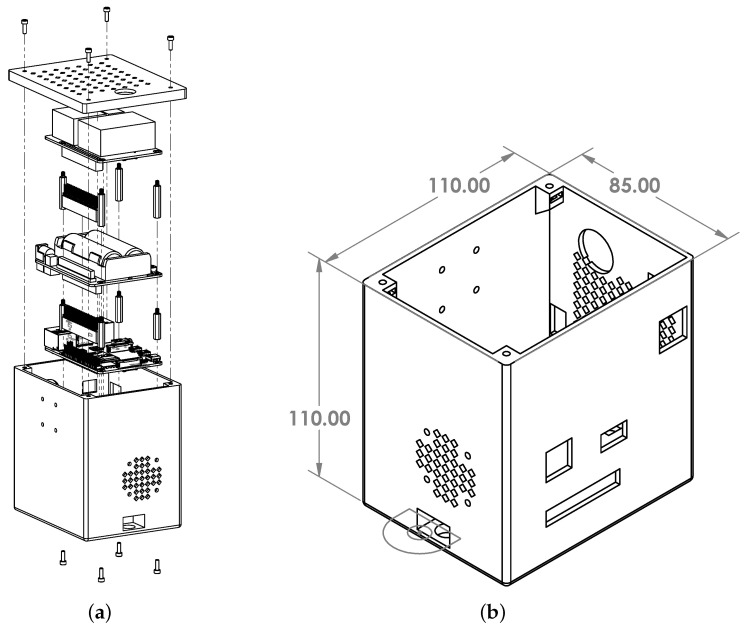
CAD designs of the assembly and enclosure. (**a**) Exploded view showing the assembly. (**b**) Isometric view of the enclosure showing cutouts for peripheral connections. The dimensions are 110 mm × 110 mm × 85 mm.

**Figure 3 sensors-23-08622-f003:**
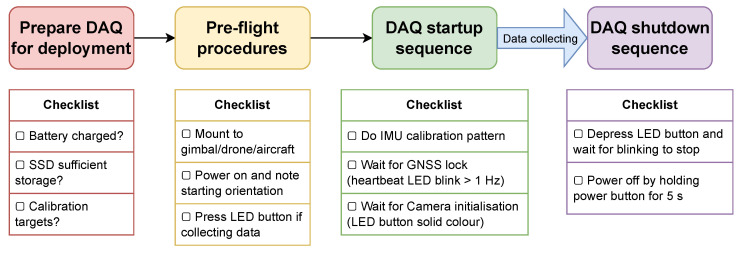
Checklists for each stage of DAQ deployment.

**Figure 4 sensors-23-08622-f004:**
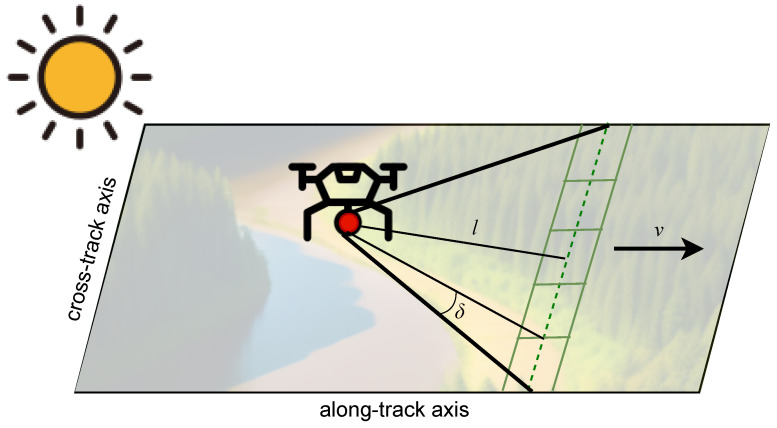
The drone payload, represented by the red disc, includes the data acquisition system and a pushbroom hyperspectral imager. As the drone flies, a swath is imaged. The forward motion provides the along-track axis. The spatial resolution is determined by the cross-track angular resolution δ and flight speed *v*.

**Figure 5 sensors-23-08622-f005:**
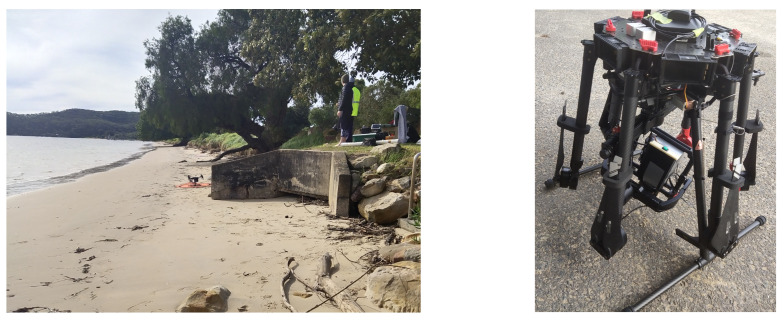
Trial site preparations. (**left**) The beach where we performed our in flight tests. (**right**) The completed DAQ and camera mounted on a gimbal ready to be deployed. The GNSS antenna was carefully routed to the top.

**Figure 6 sensors-23-08622-f006:**
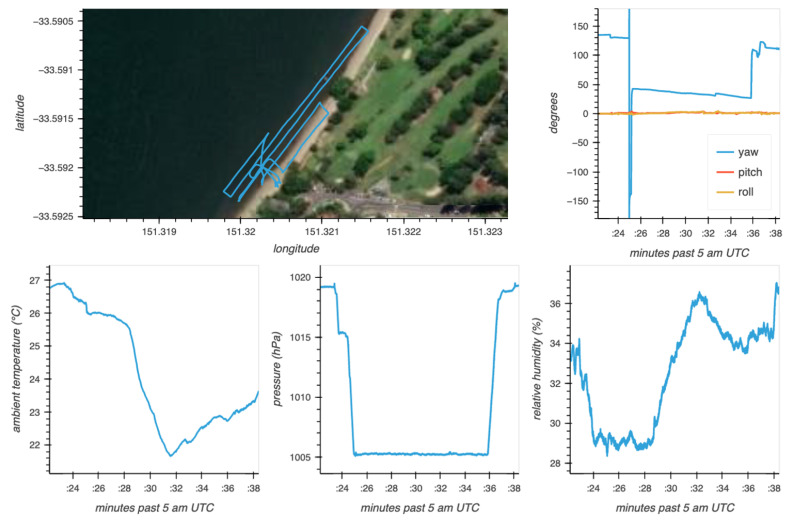
Summary plots of the navigation data showing the flight path, camera orientation in NED, and some atmospheric parameters. GNSS data were collected at 1 Hz and the rest at 100 Hz, which is faster than the camera frame rate.

**Figure 7 sensors-23-08622-f007:**
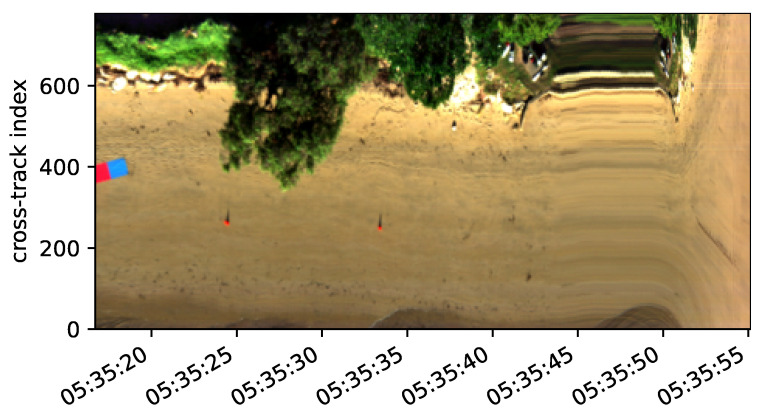
RGB composite of one hyperspectral datacube using selected bands (red = 640 nm, green 550 nm, blue 470 nm). The along-track axis is indexed by UTC time synced to the GNSS time. The stretch feature at 05:35:45–50 was due to the drone stopping before descending.

**Figure 8 sensors-23-08622-f008:**
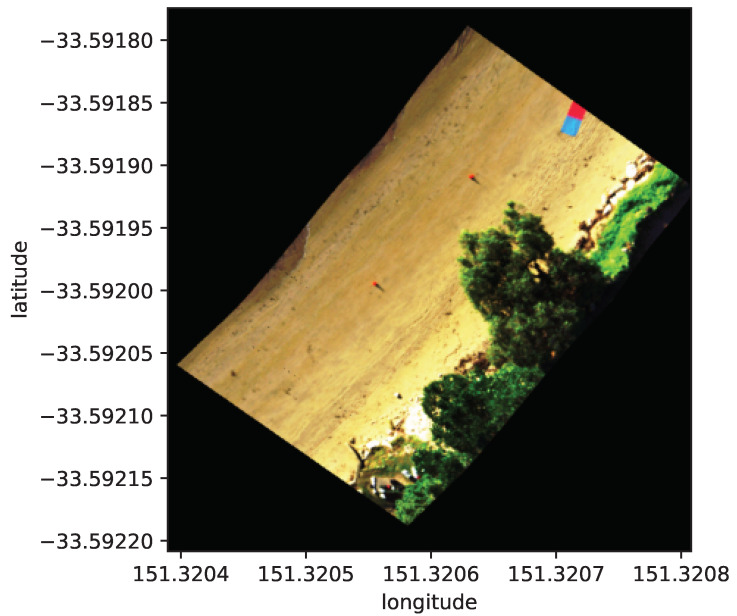
RGB composite (red = 640 nm, green 550 nm, blue 470 nm) of one hyperspectral datacube after georeferencing.

**Table 1 sensors-23-08622-t001:** Specifications for the OpenHSI camera.

Spectral range	430–830 nm
Spectral bands	213
Spatial resolution	5 cm at 120 m altitude
Cross-track pixels	800
Field of view	11${}^{\circ }$

**Table 2 sensors-23-08622-t002:** A data collection status LED and heartbeat LED indicating the operational status.

LED	Activity	Meaning
Button LED	off	Data collection not in progress
Button LED	flashing	camera being initialised or busy
Button LED	solid	data being collected
Microcontroller LED	not blinking	code hung—hit the reset
Microcontroller LED	blinking at 1 HZ	code runs as normal but missing sufficient GNSS in view
Microcontroller LED	blinking at faster than 2 Hz	Data collection with enough GNSS satellites in view

## Data Availability

Available upon reasonable request.

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
