# Peer review of "A Customisable Data Acquisition System for Open-Source Hyperspectral Imaging"

_sensors, 2023, doi:10.3390/s23208622_

Round 1

Reviewer 1 Report

The authors propose a customisable data acquisition system for hyperspectral imaging. The manuscript is clearly presented. I just wonder whether the corresponding software will be open source either to support the hardware.

n/a

Author Response

Thank you for reviewing this manuscript and for your feedback!
All the software to run this data acquisition system is open-source and can be found on our GitHub repositories (https://github.com/openhsi/hardware_files/tree/main/teensy_code and https://github.com/openhsi/openhsi/blob/master/nbs/api/sensors.ipynb).

The software library also has a documentation site for installation instructions (https://openhsi.github.io/openhsi/tutorials/installing_linux.html).

I made a few general improvements to the manuscript by adding some more details and citations. There were some grammatical inconsistencies and spelling mistakes that are now fixed. Also, the hyperspectral imager specifications are included. Lastly, I added some usage tips and future work.

Reviewer 2 Report

This paper introduces a hardware system that improves the data collection function of hyperspectral imaging, which has certain significance for improving the efficiency of remote sensing data collection. However, compared with existing UAV remote sensing data collection methods, there is no particularly impressive innovation. In addition to a relatively detailed introduction to each part of the hardware, there is no introduction to the collected remote sensing images and process method, The manuscript should be carefully revised before consideration for publication. The specific comments and questions are listed as follows:

1) The recording curve shown in Figure 7 can be provided by existing UAV flight control systems. The author should explain the purpose of recording these data using the data acquisition system of this article instead of using UAV data;

2) Spectra of select spectral targets is recorded on the left of Figure 6,  it unclear what its purpose, if it is used for data processing, the author should provide the results after using it;

3) The main purpose of this article is to provide a convenient hyperspectral data acquisition system for push-broom imaging. However, the author has not provided the resulting image after data processing using the data recorded by this device; As it's known that push-broom imaging on UAV require precise POS data for calibration,the author should provide experimental data for explanation;

4) In addition, the paper does not provide any introduction to the hyperspectral camera used in hardware as shown in fig.1, does the data acquisition system developed for certain camera or any camera, the author should provide relevant experiments to illustrate if the system can be used for several kind cameras;

Author Response

Thank you very much for reviewing this manuscript and for your feedback!

1) It is true that there is no impressive innovation compared to data collection in a UAV. However, there are benefits to having a self-contained solution and a dedicated data acquisition system. While there are commercial data acquisition (DAQ) systems designed for hyperspectral imagers, they do not work with custom built hyperspectral imagers like the one we built. In the literature, many applications of these custom built hyperspectral imagers never leave the lab because of how laborious creating and testing a DAQ is. So we offer an open-source solution.

2) The spectra of select spectral targets are only to prove that we can time synchronise hyperspectral data and is not from a RGB camera.

3) I removed the useless figure from point (2) and instead include a picture of the georeferenced data using the data we collected.

4) I added detailed on the hyperspectral camera used. The software is compatible with several detector brands and we have tested it hyperspectral cameras made with detectors from XIMEA, FLIR, and LUCID.

I made a few general improvements to the manuscript by adding some more details and citations. There were some grammatical inconsistencies and spelling mistakes that are now fixed. Lastly, I added some usage tips and future work.

Reviewer 3 Report

The authors of the research paper entitled “A customisable data acquisition system for open-source hyperspectral imaging,” developed a low-cost, open-source data acquisition system for hyperspectral imaging, addressing the current gap in the market for deployable hyperspectral imagers. Their solution collects both hyperspectral and navigation data concurrently, facilitating direct georeferencing, particularly important for remote sensing applications. They utilized commercial-off-the-shelf components, a custom-designed PCB board, and open-source software for this system. The practicality of the system was demonstrated through a drone flight test in Sydney, Australia, highlighting its potential for broad deployment in remote sensing tasks. The idea that this study is trying to convey is of good quality. Therefore, the reviewer would recommend the article for publication in the journal of Sensors provided the authors address/incorporate the following comments in the revised version of the manuscript:

Major Comments:

1-    The authors should ensure the broad relevance of their work is stressed. It is evident that they are filling a gap in hyperspectral imaging, but how does this impact future studies? How is this contribution significant compared to other imaging techniques?

2-    The authors touch upon a lot of technical components, such as the microcontroller, PCB board, Raspberry Pi 4B, Nvidia Jetson, etc. A detailed technical specification or a supplementary section detailing each component’s role and justification for its selection would strengthen the paper.

3-    While the authors emphasize the advantages of an open-source solution, a direct comparison with commercially available systems in terms of performance, accuracy, and cost would make the argument more compelling.

4-    The testing on the Matrice M600 drone is a good start, but for the system to gain broader acceptance, more extensive testing in various environments and conditions would be beneficial.

5-    While the authors briefly mention possible applications, a more extensive discussion on how this system can revolutionize current methods and provide opportunities in new areas would be valuable.

      Minor Comments:

1-    Ensure consistent terminology throughout the paper. For instance, there are different references to "imaging spectrometers" and "hyperspectral imagers." Consistency in usage will avoid confusion.

2-    It would be prudent to ensure that all referenced works are current and accurately depict the state of the art. In particular, the authors may wish to add more recent works if applicable.

3-    The testing in Sydney is briefly touched upon. Expanding on this, including challenges faced, data gathered, and insights from this specific test, would enhance this section's depth.

4-    While the conclusion briefly touches upon the weight and cost benefits, it might be valuable to discuss future upgrades, potential scalability, and other advancements they anticipate or wish to work on next.

5-    Since the hardware is open-source, it may be beneficial to also touch upon the software side of things. How easy is it to modify or adapt? Are there plans to maintain or update the software?

There are minor grammatical inconsistencies throughout the manuscript. Consider a thorough proofreading to ensure clarity and maintain a professional tone.

Author Response

Thank you very much for reviewing this manuscript and for your detailed feedback!

Major comments:
1) Added a paragraph to the conclusion about significance and impact. It reiterates some key points mentioned in the introduction.
2) It would certainly be useful to have a summary for all the components in the supplementary section. Rather than reiterate the details for all the important technical components in subsection “Electrical Components”, I added a table and explanatory notes in an appendix which outlines the role of each ancillary sensor and its accuracy.
3) It is difficult to give a comparison with commercial data acquisition systems with confidence. The selling point is that it works for the manufacturer’s hyperspectral imagers and nothing else. It seems the exact performance, accuracy, and cost is deliberately obfuscated. I did add a sentence to say that the cost of these data acquisition systems are commensurate with their hyperspectral imagers.
4) That is true. While not mentioned in this manuscript specifically, we have recently tested the data acquisition system on an aircraft as well, but those results are still processing. I added some commentary on testing in windier conditions and without the gimbal in “Usage tips and future work”.
5) I think I address this point in the updated conclusion.

Minor comments:
1) I have always used "imaging spectrometers" and "hyperspectral imagers” synonymously. However, I get your point and now consistently use “hyperspectral imagers”.
2) While there are many new works in custom built hyperspectral imager design (which we also developed), these are all tethered or lab based and do not have a data acquisition system for the purposes of direct georeferencing. I only found one recent work that used these custom hyperspectral imagers for remote sensing but it’s not open-source. I added brief commentary on the development of these recent hyperspectral imagers.
3&4) Added a subsection on “Usage tips and future work”
5) Software is also open-source and I now make some references to this. We have an extensive documentation site for it (https://openhsi.github.io/openhsi/). This can be found in one of the website citations. While I don’t mention plans for maintenance in the manuscript, there are other people in our research group besides myself (first author) who are using and extending the software.

There were indeed some grammatical inconsistencies and spelling mistakes that are now fixed.
Again, thank you so much for your helpful suggestions.

Round 2

Reviewer 2 Report

The innovation of this paper seems to emphasize that the data acquisition system developed can provide the camera with position and attitude information to correct the deformation during the imaging process of the push-broom camera, but it is doubtful whether the low-end hardware equipment can really solve the problem in reality. It is suggested that the author give more original and processed hyperspectral image results obtained by using this system to enhance the persuasiveness of the paper.

Author Response

Thanks for the feedback and we shared the same concerns when choosing components. During development, we found that these sensors had random noise much lower than the stated manufacturer accuracy. As in, if we sit the IMU on the desk, it will stay constant rather than jump around by 1 degree. The same with the GNSS module except it would slowly drift over the course of minutes. Since the timescale of our data collection happened within a few tens of seconds, these long-term drift errors are not apparent in our georeferenced data presented in the manuscript.

We plan to test the data acquisition system in more challenging situations. The data presented here was our first successful attempt. We had very light winds (<1 m/s) on the day, and we used a gimbal for stabilisation which contributed to high quality data. For example, what would the performance look like if we did not use a gimbal? What if there was a lot of turbulence? What if the sensors were very noisy?

Short of running another field trial, which is not possible as I am overseas right now and it takes time to organise, let me demonstrate the georeferencing performance when I manually add random noise to the position and orientation data according to the stated manufacturer performance. The effect of noise is visually compared in Fig A1 and A2. In reality, one would filter this random noise (depending on the frequency of the dynamics) so the effect will not be as exaggerated as in Fig A2.

Fig A1: Georeferenced data using collected position and orientation data

Fig A2: Georeferenced data after adding random noise (orientation ± 1°, position ± 1.5 m) to position and orientation data. No filtering of the data.  

While low end, these sensors are readily available, affordable, and offer good performance suitable for direct georeferencing. I added a paragraph about this in Appendix B. 

Reviewer 3 Report

The authors of the manuscript entitled “A customisable data acquisition system for open-source hyperspectral imaging” have improved the quality of the paper in terms of organization and writing. In the reviewer’s opinion, the current manuscript can be accepted for publication on Sensors. 

Author Response

Thank you for your suggestions. I added a short paragraph in Appendix B commenting on the manufacturer's stated accuracy and our georeferencing performance.